# Improving semi-arid agroecosystem services with cover crop mixes

**Elizabeth A. Moore**[1], **Urszula Norton**[1,2] *

**1** Department of Plant Sciences, Department 3354, University of Wyoming, Laramie, WY, United States of America, **2** Program in Ecology, University of Wyoming, Laramie, WY, United States of America

* unorton@uwyo.edu

**Data Availability Statement:** University of Wyoming WyoScholar data repository. https://doi.org/10.15786/3d2r-ew09.

**Funding:** UN USDA -NIFA 2019-51300-30476 This research was funded by the United States Department of Agriculture (USDA) through the

## Abstract

Winter wheat (*Triticum aestivum*, L.) production in the semi-arid US Northern High Plains (NHP) is challenged by frequent droughts and water-limited, low fertility soils. Composted cattle manure (compost) and cover crops (CC) are known to provide agroecosystem services such as improved soil health, and in the CC case, increased plant diversity, and competition with weedy species. The main concern of planting CC in winter wheat fallow rotation in regions that are more productive than the NHP, however, is the soil moisture depletion. It is unknown however, whether addition of CC to compost-amended soils in the NHP will improve soil properties and agroecosystem health without compromising already low soil water content. The main objective of this study was to assess the effects of four CC treatments amended with compost (45 Mg ha$^{-1}$) or inorganic fertilizer (IF) (.09 Mg ha$^{-1}$ mono-ammonium phosphate, 11-52-0 and 1.2 Mg ha$^{-1}$ ammonium sulfate, 21-0-0) on the presence of weeds, soil and plant total carbon (C), nitrogen (N), and biological dinitrogen (N$_2$) fixation (BNF). Mycorrhizal Mix (MM), Nitrogen Fixer Mix (NF), Soil Building Mix (SB), a monoculture of phacelia (*Phacelia tanacetifolia* Benth L.) (PH), and a no CC control (no CC) were grown in native soil kept at 7% soil moisture in a greenhouse for a period of nine weeks. When amended with compost, MM was the most beneficial (48 g m$^{-2}$ BNF and 1.7% soil C increase). SB had the highest germination, aboveground biomass, and decreased weed biomass by 60%. It also demonstrated the second highest amount of BNF (40 g m$^{-2}$) and soil C increase by 1.5%. On contrary, IF hindered BNF by almost 70% in all legume-containing CC treatments and reduced soil C by 15%.

## Introduction

The inclusion of cover crops (CC) in crop rotations have been successfully practiced worldwide, yet the CC adoption in cold, semi-arid environments, such as the US Northern High Plains (NHP), is very limited. When planted during the "off- crop season," CC can help build up soil organic matter (SOM), improve soil aggregation, prevent erosion, and reduce nutrient loss [1]. In addition to soil benefits, CC also provide effective weed biomass reduction [2]. There are concerns however, that CC draws down soil moisture and compromise winter wheat (*Triticum aestivum*, L.) yield [3].

National Institute of Food and Agriculture (NIFA) gran number 2019-51300-30476. The funders did not play any role in the study design, data collection and analysis, decision to publish, or preparation of the manuscript.

**Competing interests:** The authors have declared that no competing interests exist.

The NHP has a growing area under organic winter wheat production and active soil health and weed management strategies are needed to help producers maintain their certification. The most common CC mix recommended locally is field pea (*Pisum sativum* L.) mixed with oat (*Avena sativa* L.). Exploring alternative CC mixes that help fulfill production goals and aid in management objectives are, however, much needed as this mix often experiences very poor establishment.

The term "cover crops" refers to monocultures or mixes of two or more plant species with various traits. Species from plant families such as *Poaceae*, *Fabaceae*, *Brassicaceae* and *Asteraceae* are commonly used in CC mixes. Each family has beneficial characteristics for agroecosystems. Fibrous roots of plants from the *Poaceae* family are known to stabilize soil, capture nutrients, and produce high biomass that returns organic matter (OM) to the ground upon termination [4]. Species from the *Fabaceae* family live in symbiosis with bacteria capable of atmospheric dinitrogen ($N_2$) fixation. This relationship supplies nitrogen (N) to the plants during the growing season. Upon plant senescence, N in plant biomass is deposited in the soil [5]. Species from the *Brassicaceae* family put out deep taproots that help alleviate soil compaction. Seeds are fast to germinate, and hence, have a time advantage to establish themselves ahead of weeds [6]. Sunflower (*Helianthus annuus*) from the *Asteraceae* family is used in CC mixes to improve soil hydraulic properties. Sunflower roots access water deeper in soil layers and redistribute it near the soil surface to benefit shallow rooted species in a CC mix [7].

Comparing $^{15}N$ natural abundance ($\delta^{15}N$) between a soil, plant, inorganic fertilizer or compost provides many insights on the pathways of plant N uptake and soil organic matter decomposition (He et al., 2009). This approach is based on the premise that the majority of naturally occurring N is in the $^{14}N$ form (99.6%) and the remaining 0.4% of N exists as $^{15}N$ [8]. The ratio of these two isotopes ($^{15}N/^{14}N$) in the atmosphere is constant at .0036765 and is used as a reference. Many microbial and plant physiological processes discriminate against heavier $^{15}N$ compared to $^{14}N$ and can indicate what forms of N are available to the plant [9]. For example, during the composting process, microbes use the $^{14}N$ resulting in greater compost $^{15}N$ enrichment. When inorganic fertilizer is manufactured from atmospheric N, the $^{15}N$ level is of the fertilizer is the same as the atmosphere, which is close to zero. As cover crops and weeds compete for the N provided from these inputs, plant tissue $^{15}N$ concentrations will increase or decrease accordingly. These two inputs with very different $^{15}N$ concentrations provide a traceable N source and observations on cover crop versus weedy species competition can be made.

During atmospheric $N_2$ fixation by bacteria living on leguminous roots, the soil bacteria discriminate against $^{15}N$ as they only utilize the lighter isotope $^{14}N$. This discrimination ratio is transferred to the legume tissue and provides a traceable N source when observing legume plant biological N fixation BNF [8,9].

Soil amendments, like composted cattle manure (compost) and inorganic fertilizer (IF) are used to boost cash crop performance and yield [10,11] but how these soil amendments impact CC performance has not been researched. The main objective of this project was to observe how CC mixtures and weedy species interact with, respond to, and compete for soil N sourced from soil amendments under low water conditions.

## Materials and methods

### Site description

This study was carried out for a period of nine weeks in the spring of 2019 at the University of Wyoming Laramie Research and Education Center (LREC) Greenhouse in Laramie, Wyoming (41.31˚N, 105.59˚, 2214 meters above sea level). This length of time was chosen arbitrarily to mimic the short window of time, during which cover crops are planted, produce

**Table 1. Texture, pH, electrical conductivity (EC), bulk density, δ $^{15}$N, total nitrogen (Total N), total organic carbon (OC), inorganic C (IC), C to N ratio, potentially mineralizable nitrogen (PMN) and available phosphorus (Available P) in soil and compost.**

| Parameter | Soil | Compost |
|---|---|---|
| Texture | Silt Loam | Fibrous |
| pH (1:2 soil: water) | 7.80 | 8.46 |
| EC (μS cm$^{-1}$) (1:1 soil: water) | 315 | 2870 |
| Bulk Density (g cm$^{-3}$)(dry weight) | 1.40 | 0.98 |
| δ $^{15}$N (‰) | 4.91 | 10.4 |
| Total N (g kg$^{-1}$) | 1.60 | 12.4 |
| OC (g kg$^{-1}$) | 12.0 | 85.7 |
| IC (g kg$^{-1}$) | 4.5 | 9.4 |
| C to N ratio | 10 | 7 |
| PMN (mg kg$^{-1}$) | 16.7 | 68.2 |
| Available P (mg kg$^{-1}$) | 23.5 | 36.2 |

biomass, flower and are either harvested or terminated, depending on the production goals. The greenhouse facility was not equipped with any artificial light and the average light intensity averaged 23,497.9 lumens m$^{-2}$ [12]. Day and night temperatures were held constant at 21°C and 18°C, respectively. At the beginning of the experiment, the day length was 11 hours and 35 minutes and gradually increased to 14 hours and 17 minutes by the end of the experiment [13].

Soil was sourced from the fallow phase of the 15-year long winter wheat -fallow rotation located at the James C. Hageman Sustainable Agriculture Research and Extension Center (SAREC) in Lingle, WY (42.14°N, 104.35°W, and 1272 meters above sea level). The area has a semi-arid climate with 125 frost-free days. Average annual precipitation ranges between 300 to 400 mm and average high and low temperatures are 15.2°C and 0°C, respectively (35-year averages) [11]. Soils are loamy, mixed, active, mesic Ustic Torriorthents with <1% soil organic matter (SOM) and a maximum 3% slope [11,14]. The top 10 cm of surface soil was collected with a hand trowel. Thirty individual sub-samples were placed in a bucket, homogenized by hand, and coarse rock fragments and visible roots removed. Soil was transported to the lab, air dried and sieved through a 2-mm sieve [15].

After drying and sieving, the soil was analyzed for chemical properties. Soil pH was slightly alkaline (pH of 7.8), electrical conductivity EC was low (315 μS cm$^{-1}$) and bulk density was 1.40 g cm$^{-3}$ (Table 1). The isotopic signature was 4.91 ‰ δ $^{15}$N (Table 1). Total N was 1.60 g kg$^{-1}$, organic carbon (OC) was 12.0 g kg$^{-1}$, inorganic carbon (IC) was 4.5 g kg$^{-1}$, C to N ratio of 10, potentially mineralizable N was 16.7 mg kg$^{-1}$ and available phosphorus (Available P) was 23.5 mg kg$^{-1}$.

## Experiment set up

Sixty plastic potting containers (volume of 2048 ml) were filled with approximately 1500 ml of soil. Twenty pots were amended with 72.6 g (dry bases) of composted cattle manure (compost). This amount is equals to a rate of 45 dry tons ha$^{-1}$ and is based on recommendations from earlier studies that showed 11% release of N from compost in year one [11]. Compost contained 1.24% N, 8.57% C, 0.85% available P, 0.89% iron (Fe), and had a C-to-N ratio of 7 (Table 2). Compost addition contributed 558 kg ha$^{-1}$ total N and 382.5 kg ha$^{-1}$ P. Compost natural abundance of N (δ$^{15}$N) isotope was 10.4‰.

**Table 2. Concentrations (%) of macronutrients, micronutrients, and trace elements of Composted cattle manure (total organic nitrogen (Total N), ammonium, nitrate, available phosphorus, potassium, calcium, magnesium, sulfur, boron, copper, iron, manganese, zinc, and sodium.**

| Chemical Composition | Compost |
|---|---|
| **Macronutrients:** | -%- |
| Total N | 1.24 |
| Ammonium | 0.004 |
| Nitrate | 0.086 |
| Available Phosphorus | 0.85 |
| Potassium | 2.13 |
| Calcium | 3.93 |
| Magnesium | 0.68 |
| Sulfur | 0.41 |
| **Micronutrients:** | -%- |
| Boron | .003 |
| Copper | .002 |
| Iron | .89 |
| Manganese | .02 |
| Zinc | .01 |
| **Trace Elements:** | -%- |
| Sodium | 3.30 |

Twenty pots were amended with inorganic fertilizer (Inorganic Fertilizer) at rates comparable to what local producers use and was comprised of mono-ammonium phosphate ($NH_4H_2PO_4$) or MAP (11-52-0), at a rate of .09 Mg ha$^{-1}$ and ammonium sulfate [$NH_4(SO_4)_2$] (21-0-0) at a rate of 1.2 Mg ha$^{-1}$. The composition of the inorganic fertilizers are as follows: $NH_4H_2PO_4$ contains on average, 11% Nitrogen (N), 52% Phosphorus (P), and 0.6% Fluoride (F) and has $\delta^{15}N$ of 1.7‰ [16]; while [$NH_4(SO_4)_2$] contains 21% N and 24% Sulfur (S) and has $\delta^{15}N$ of -0.9‰ [16]. In all, the Inorganic Fertilizer treatment enriched soil with 351 kg ha$^{-1}$ N, 207 kg ha$^{-1}$ P, 288 kg ha$^{-1}$ S, and 5.4 kg ha$^{-1}$ F. The inorganic fertilizer treatment provided a comparison to compost to assess realized benefits of compost added at rates local producers apply. Inorganic fertilizer also acted as a proxy for readily available plant nutrients that compost provides. In addition to the compost treatment and inorganic fertilizer treatment, there was a control treatment with no soil additives (Control).

Cover crop mixes were formulated to perform and deliver specific management goals most suitable for the local weather and soil conditions. Five CC treatments were composed of legumes, broadleaves, and grasses(Table 3). The following CC treatments were planted at a low seeding rate as recommended by the cover crop vendor (Table 4) [16]: (1) phacelia monoculture (*Phacelia tanacetifolia* Benth. L.) (PH); (2) soil building mix (SB), nine species total (four legumes, one brassica, two broadleaf and two grasses); (3) nitrogen fixing mix (NF), ten species total (six legumes, one brassica, two broadleaf and one grass); (4) mycorrhizal mix (MM), 14 species total (five legumes, four broadleaf and five grasses) and (5), no cover crop control (CON).

Phacelia is a native annual forb from the *Hydrophyllaceae* family [17]. It is fast germinating large broadleaf biomass producing form effective in competitive N uptake [18]. Purple flowers attract beneficial insects such as hover flies (*Syrphidae*), that prey on aphids (*Aphidoidea*) [19] and is very pollinator friendly [17].

Prior to planting, pots were watered uniformly with 105 ml of purified water to adjust soil water content to 7% of water filled pore space (WFPS), which represents the normal soil water

**Table 3. Common name, scientific name, and plant family for phacelia (PH), soil building mix (SB), nitrogen fixing mix (NF), mycorrhizal mix (MM) treatments.**

| Common Name | Scientific Name | Plant Family | PH | SB | NF | MM |
|---|---|---|---|---|---|---|
| Chickpea | *Cicer arientinum* | Legume | | | X | |
| Spring Pea | *Pisum sativum* | Legume | | X | X | |
| Spring Lentil | *Lens culinaris* | Legume | | X | X | X |
| Chickling Vetch | *Lathyrus sativus* | Legume | | | X | |
| Common Vetch | *Vicia sativa* | Legume | | X | X | X |
| Berseem Clover | *Trifolium alexandrinum* | Legume | | | | X |
| Crimson Clover | *Trifolium incarnatum* | Legume | | X | X | |
| Persian Clover | *Trifolium resupinatum* | Legume | | | | X |
| Mung Bean | *Vigna radiata* | Legume | | | | X |
| Rapeseed | *Brassica napus* | Broadleaf | | X | X | |
| Sunflower | *Helianthus annus* | Broadleaf | | X | X | X |
| Flax | *Linum usitatissimum* | Broadleaf | | X | X | X |
| Phacelia | *Phacelia tanacetifolia* | Broadleaf | X | | | X |
| Safflower | *Carthamus tinctorius* | Broadleaf | | | | X |
| Barley | *Hordeum vulgare* | Grass | | X | | X |
| Oats | *Avena sativa* | Grass | | X | X | X |
| White Wonder Millet | *Setaria italica* | Grass | | | | X |
| Proso Millet | *Panicum miliaceum* | Grass | | | | X |
| Brown Top Millet | *Urochloa ramosa* | Grass | | | | X |

content for the field growing season [11]. Pots were transferred to a greenhouse, arranged in a completely randomized design with four replications on a north-south longitudinal bench (Fig 1). Pots were rearranged weekly and soil moisture was adjusted daily to maintain constant 7% WFPS. This was done by comparing individual pot weights to a reference vegetation-free pot with soil kept at 7% WFPS (Fig 1). Based on daily soil water evaporation loss from the reference pot, all experimental pots were supplied with equal amounts water.

## Plant and soil analyses

The experiment was terminated after nine weeks. Aboveground plant biomass was clipped and separated to CC and weedy species. All individual plants were identified and counted. Wet weights were recorded using Ohaus Scout Pro SP402 portable digital scale (Parsippany, New Jersey, USA). Aboveground biomass, separated by cover crops and weedy species, were stored in paper bags, oven-dried at 65˚C for 48 hours, and dry weights recorded.

Soil and roots from each pot were emptied to a 2-mm sieve to separate roots and coarse fragments. Soil was homogenized and placed in a plastic zipper bag. All samples were kept in a cooler until laboratory analyses performed within 36 hours of collection. Soil pH and EC were

**Table 4. Total number of species, number of leguminous species, grass to broadleaf ratio and planting density for phacelia (PH), soil building mix (SB), nitrogen fixing mix (NF) and mycorrhizal mix (MM) treatments.**

| Cover Crop Treatment | Total Species | Leguminous Species | Grass to Broadleaf Ratio | Planting Density (kg ha$^{-1}$) |
|---|---|---|---|---|
| PH | 1 | 0 | 0:100 | 6.7 |
| SB | 9 | 4 | 60:40 | 44.8 |
| NF | 10 | 6 | 25:75 | 50.4 |
| MM | 14 | 5 | 50:50 | 50.4 |

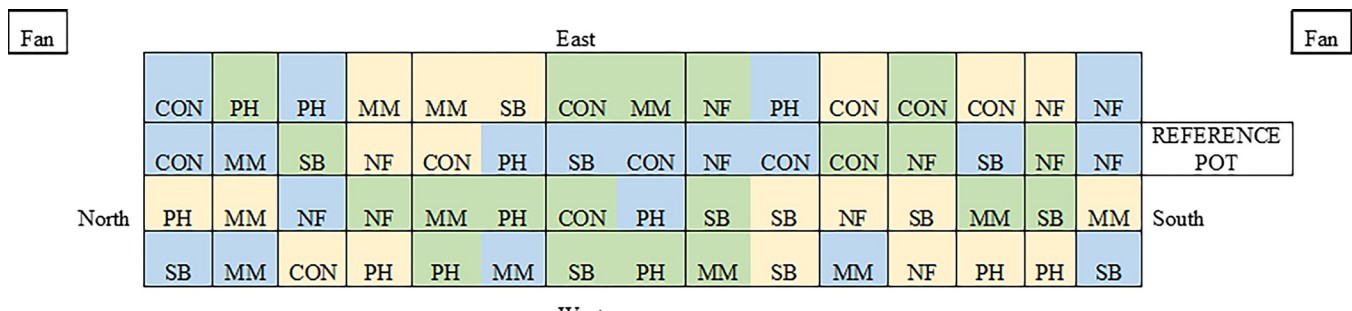

**Fig 1. Greenhouse experimental layout.** Treatments include control (CON), phacelia (PH), soil building mix (SB), nitrogen fixer mix (NF) and mycorrhizal mix (MM). Soil additives include no soil additive (yellow), composted cattle manure (green) and inorganic fertilizer (blue). Boxes labeled "fans" represent cooling fans, box labeled "reference pot" represents pot filled with the same soil but no plants used as a reference for soil water adjustments. Pots were rotated weekly.

tested using an Oakton 2700 series benchtop meter at a 1:2 soil-to-water ratio. Inorganic N was determined by adding 10 g of fresh soil to 25 ml of two molar potassium chloride (2 $M$ KCl), placed on a shaker for 30 minutes, and filtered through ash-free filter paper (Q5 Fischer Scientific, USA). The extract was then analyzed on a spectrophotometer microplate reader (UV-VIS Biotek Instruments, Highland park, USA) for ammonium ($NH_4$) using sodium salicylate (Reagent A) and 2% bleach mixed with 1.5 $M$ sodium hydroxide (Reagent B) [20]. Nitrate was analyzed using the same procedure with vanadium chloride [21] as the sole reagent. Both parameters were then combined for a single estimate of inorganic N.

Potentially mineralizable nitrogen (PMN) was assessed by incubating five grams of fresh soil in 12.5 mL of DI water for 14 days at 25°C to create an anaerobic environment [22]. Oxygen present in the headspace of the centrifuge tube was flushed with dinitrogen ($N_2$) gas prior to sealing the tube. After two weeks, 12.5 ml of 4 $M$ KCl was added, shaken for 30 minutes, stored at 4°C overnight, and filtered through ash-free filter paper (Q5 Fischer Scientific, USA). The resulting extract was then analyzed using a spectrophotometer microplate reader (UV-VIS Biotek Instruments, Highland Park, USA), with PMN calculated as the difference between pre- and post-incubation $NH_4$ concentrations.

Dissolved organic carbon (DOC) concentrations were analyzed using the Newcomb-Carrillo method [23], with a 1:2.5 soil: $K_2SO_4$ (0.5 $M$) extracts that were shaken for 30 minutes, stored at 4°C overnight, and filtered through ashless filter paper (Q5 Fisher Scientific, USA). Samples were analyzed on a total organic carbon (TOC) analyzer (Shimadzu TOC-VCPH with TNM-1, Japan). To calculate soil inorganic carbon (IC), a bicarbonate extraction was completed using 0.5g of air-dried, finely ground soil placed in glass medicine bottles. Two mL of 6$M$ $FeCl_2$ was pipetted into small glass cylinders, then dropped into larger glass medicine bottles, which were sealed with a rubber septum and tightly crimped. The bottles were rigorously shaken, then allowed to rest for six hours before testing the bottles using a pressure calcimeter (Serta, 280E, USA). Soil IC was then estimated with the Sherrod method [24]. Soil organic carbon was calculated as the difference of soil IC and total C.

## Nitrogen isotopes

Soil, compost, and plant biomass material were finely ground using a Wiley Mill (Wiley Laboratory Mill, Model 4, Arthur H. Thomas Co., Philadelphia, USA) to pass through a 1-mm screen. Soil and compost samples (25–26 mg) and plant material (2.5–3 mg) were wrapped in tin capsules and analyzed for total carbon (TC), total nitrogen (TN), and stable nitrogen isotope ratio ($^{15}N/^{14}N$) on an isotope mass spectrometer (Costech 4010 Elemental Analyzer

Thermo Delta Plus XP IRMS). Reproducibility as determined through replicate measurements was better than 0.001‰ for $\delta^{15}N$ [25]. Measurements were normalized based on the measured values of standards material (Glutamic acid, $\delta^{15}N_{Air}$: 5.8‰ ± .15‰), relative to Vienna Pee Dee Belemnite (VPDB) and air, respectively, where:

$$\delta^{15}N(‰) = [(R_{sample} - R_{air})/R_{air}] \times 1000, \text{ where } R = {}^{15}N/{}^{14}N$$

The $\delta^{15}N$ (‰) were used to determine biological N fixation (BNF). As discussed above, atmospheric N fixation by bacteria living on leguminous roots provides a traceable N source [8]. An average of the weedy control treatment $\delta^{15}N$ (‰) was used as a proxy for BNF in treatments absent of legumes. This average $\delta^{15}N$ (‰) was subtracted from each individual CC treatment $\delta^{15}N$ (‰) to establish the $\delta^{15}N$ (‰) from BNF, which was then divided by ten to convert per mil to percent. This percent was then multiplied by the treatment biomass to obtain biomass from BNF.

$$\text{CC BNF} = ((\text{Average } \delta^{15}N \text{ Weedy Control} - \delta^{15}N \text{ CC Trt})/10) * \text{BM CC Trt}$$

Where:
BM = Cover Crop Biomass

$$\text{Weeds BNF} = ((\text{Average } \delta^{15}N \text{ Weedy Control} - \delta^{15}N \text{ Weeds in CC Trt})/10) * \text{BM Weeds in CC Trt}$$

Where:
BM = Weed Biomass in CC Trt

## Statistical analyses

The experiment was set up in a completely randomized design with four replications. Soil moisture, soil organic C, plant biomass, total percent C in plant biomass, total percent N in plant biomass, biological nitrogen fixation, C:N ratio of plant biomass, $\delta^{15}N$ (‰) in plant biomass were analyzed with R version 3.6.2 [26]. Cover crop treatment and soil additive treatment (compost and inorganic fertilizer) were fixed effects. Data were tested for normality using the Shapiro–Wilk test. Soil moisture, plant biomass, total percent Carbon, total percent Nitrogen, and biological nitrogen fixation were square root transformed to meet assumptions of normality and means back transformed for reporting. Carbon to Nitrogen ratio, Nitrogen isotope and soil organic carbon were normal and analyzed without transformations. Soil gravimetric and chemical characteristics, and stable isotope concentrations were assessed using two-way Analysis of Variance (ANOVA) with significance at a minimum of P ≤ 0.05. No interactions between CC treatment and soil additive treatment were found for this data set. Means separations were performed using Tukey HSD at a minimum of P ≤ 0.05. Regression analyses was used to compare the relationship of weed biomass to cover crop biomass for cover crop treatments [27].

## Results

### Cover crops

Cover crop emergence was three to four times greater in NF, MM and SB than in PH (Table 5). Berseem clover and Persian clover established successfully while chickpea, field pea, spring lentil, chickling vetch and mung bean failed to germinate. Four out of five species in the NF treatment that failed to germinate were legumes and the fifth one was sunflower. Common

**Table 5. Common name and plant density (number $m^{-2}$) of emerged plants in phacelia (PH), soil building mix (SB), nitrogen fixer mix (NF), and mycorrhizal mix (MM) treatments.**

| Common Name | PH | SB | NF | MM |
|---|---|---|---|---|
| | ----number $m^{-2}$ ---- | | | |
| Chickpea | - | - | 0 | - |
| Spring Pea | - | 0 | 0 | - |
| Spring Lentil | - | - | 0 | 0 |
| Chickling Vetch | - | - | 0 | - |
| Common Vetch | - | 112 | 68 | 0 |
| Berseem Clover | - | - | - | 124 |
| Crimson Clover | - | 32 | 206 | 0 |
| Persian Clover | - | - | - | 87 |
| Mung Bean | - | - | - | 0 |
| Rapeseed | - | 74 | 81 | - |
| Sunflower | - | 62 | 0 | 0 |
| Flax | - | 124 | 62 | 93 |
| Phacelia | 242 | - | - | 149 |
| Safflower | - | - | - | 62 |
| Barley | - | 93 | - | 93 |
| Oats | - | 124 | 407 | 124 |
| White Wonder Millet | - | - | - | - |
| Proso Millet | - | - | - | 62 |
| Brown Top Millet | - | - | - | - |
| TOTAL | 242 | 621 | 824 | 794 |

vetch, crimson clover and berseem clover were the most frequently observed legumes that successfully germinated across all mixes containing legumes. As such, common vetch, flax, and oats dominated SB; oats and crimson clover dominated NF and berseem clover, phacelia, and oats dominated MM (Table 5).

All CC treatments and CON produced comparable aboveground biomass (Fig 2). Out of CC treatments, MM, NF, and SB had the greatest CC biomass compared with PH. The highest BNF was observed in control and compost amended soils. Soil in PH treatment had negligible BNF (Fig 3).

## Weeds

All CC treatments reduced weed biomass 50% to 60% (Fig 2). Through linear regression analysis, no significant relationship of weed biomass to CC biomass was found for PH (slope = -0.43) and MM (slope = -0.37), but a significant relationship was found for SB (slope = -0.44) and NF (-0.592); F(4, 44) = 0.0001. A Tukey pairs comparison revealed that SB (slope -0.44) and NF (-0.592), p = 0.98 were not significantly different. The rate of weed biomass reduction was estimated at a rate of Weed biomass = -0.59(NF CC biomass) + 142.01 ($R^2$ = 0.54) in NF and Weed biomass = -0.44(SB CC biomass) + 138.66 ($R^2$ = 0.55) in SB. As cover crop biomass in NF and SB increased, there was a relational rate of decrease in weed biomass (Table 6).

## Carbon and nitrogen

Cover crops in SB and NF mixes had the highest plant tissue total C while CC in PH and CON had the lowest (Table 7). There were no observable differences in total N in the CC tissue. The resulting highest C:N ratio was in cover crops in SB mix (Table 7). In contrast, weeds had

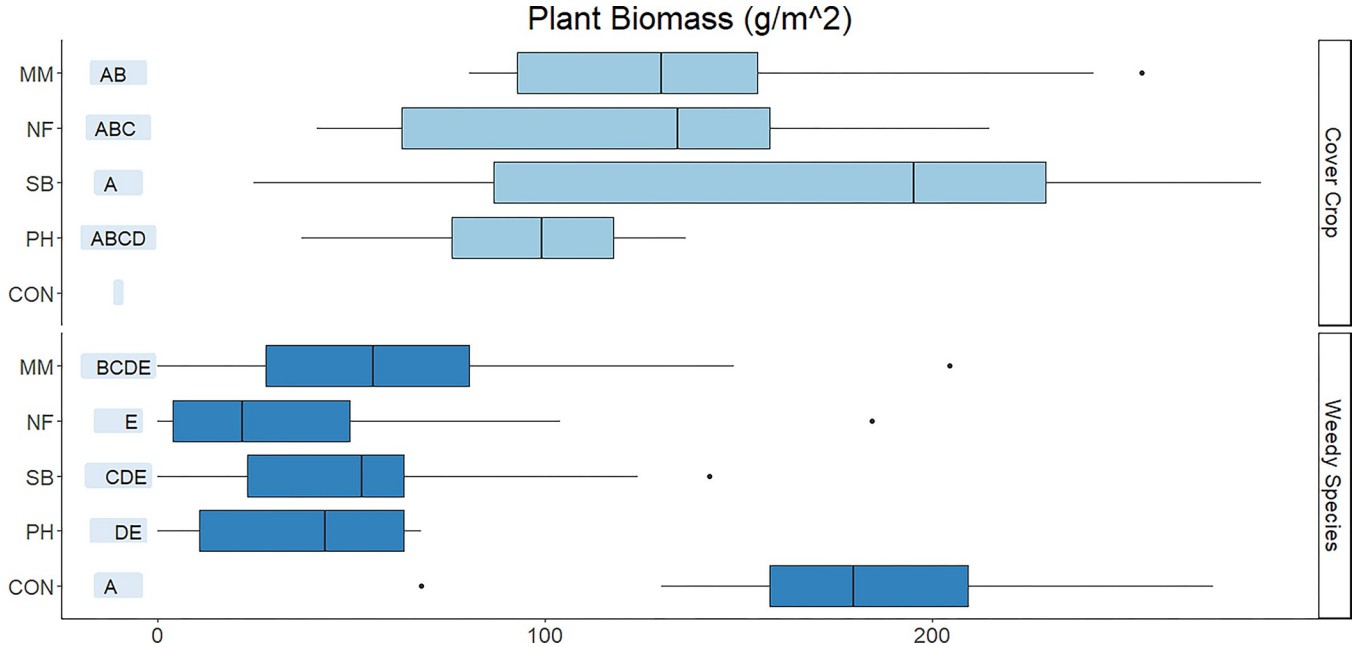

**Fig 2. Cover crop and weedy species aboveground plant biomass in control (CON), phacelia (PH), soil building mix (SB), nitrogen fixing mix (NF), and mycorrhizal mix (MM) treatments.** Upper-case letters demonstrate treatments differences at p ≤ .05.

comparable tissue total C across all CC treatments. Tissue total N was comparable across all CC treatments than CON, which turned significantly higher than tissue total N in cover crops in MM mix (Table 7). Weed tissue C:N ratio was also the highest in MM and the lowest in CON and PH.

Cover crops planted in amended soils showed differential rates of BNF. In general, BNF ranged between 2 to 40 g m$^{-2}$ in the control, 10 to 50 g m$^{-2}$ in the Compost treatment and 1 to 15 g m$^{-2}$ in the Inorganic Fertilizer treatment (Fig 3C). For CC, the highest BNF values were in MM in compost (Fig 3B) followed by SB in control (Fig 3A). Phacelia had the lowest overall values of BNF across all treatments ranging from 0 to 9 g/m$^2$. The lowest overall values were observed in IF treatment (Fig 3C).

Soil δ$^{15}$N were 4.91 ‰ in the control treatment, 5.93 ‰ in the Compost treatment, and 5.09 ‰ in the Inorganic Fertilizer treatment (Fig 4). These soil δ$^{15}$N (‰) act as a baseline to which the weedy species δ$^{15}$N (‰) and CC plant tissue δ$^{15}$N (‰) can be compared. In the control treatment, weedy species δ$^{15}$N (‰), and PH δ$^{15}$N (‰) hovered around the soil δ$^{15}$N line (Fig 4A), whereas NF mix δ$^{15}$N (‰), SB mix δ$^{15}$N (‰) and MM mix δ$^{15}$N (‰) and the PH weedy species δ$^{15}$N (‰) were lower than soil δ$^{15}$N (‰) (Fig 4A). In the Compost treatment, the δ$^{15}$N of all observations increased (Fig 4B). Weedy species δ$^{15}$N observations in NF, SB, and no cover crop were not different than compost δ$^{15}$N (10.4 ‰). All CC δ$^{15}$N (‰) were lower than compost δ$^{15}$N (‰) and NF mix δ$^{15}$N (‰) was lower than the soil δ$^{15}$N (Fig 4B). In the Inorganic Fertilizer treatment, all CC treatments δ$^{15}$N (‰) were not different than the soil δ$^{15}$N (5.09 ‰), except NF mix (Fig 4C). All weedy treatments δ$^{15}$N (‰) were lower than the soil δ$^{15}$N (‰) except for weedy species δ$^{15}$N (‰) in the no CC treatment (Fig 4C).

Soil organic C ranged between 1.10% and 1.75% (Fig 5). In native soil, without a soil additive, SB had the highest SOC but was only significantly different than CON (Fig 5A). In the Compost treatment, the highest SOC concentrations were in SB or MM and were significantly

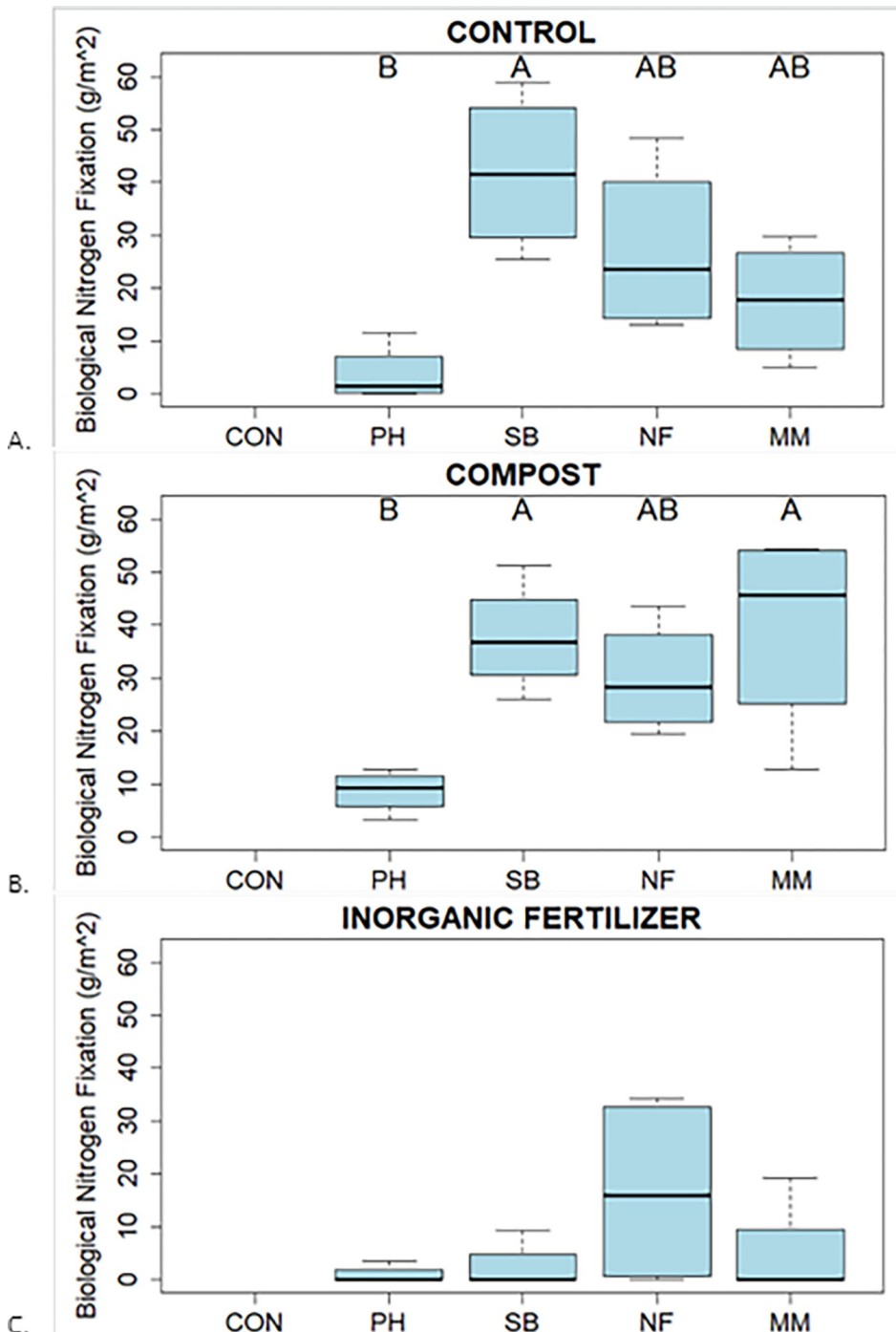

**Fig 3. Biological nitrogen fixation (BNF) for cover crop treatments** (no cover crop (CON), Phacelia (PH), Soil building mix (SB), Nitrogen fixer mix (NF) and Mycorrhizal mix (MM)) in control (A), compost (B), and inorganic fertilizer (C). BNF was calculated using weedy species of control treatment for each soil treatment as a reference. Different upper-case letters are different across cc treatments at p ≤ .05 and provide a stepwise statistical comparison of each parameter by cover crop mix treatment.

**Table 6. Regression analysis of weed biomass (g m$^{-2}$) by cover crop biomass (g m$^{-2}$) for nitrogen fixer mix (NF) and soil building mix (SB).** $R^2$ followed by one star indicates significance at p = .01 level.

| Treatment | Regression | R$^2$ |
|---|---|---|
| NF | Weed biomass = -0.59(NF CC biomass) + 142.01 | 0.54 * |
| SB | Weed biomass = -0.44(SB CC biomass) + 138.66 | 0.55 * |

different than NF. Percent SOC was lower when Inorganic fertilizer was applied but among CC treatments, no significant differences were found.

## Discussion

The CC mixes used in this study were formulated to perform and deliver specific management goals. Some of the CC mixes included diverse species within the same plant family but not all those species were present at termination. Soil building mix had the fewest species among CC mixes and the lowest planting density but outperformed the other mixes among soil and plant parameters. When designing a CC mix, selection of species from the *Poaceae*, *Fabaceae*, and *Brassicaceae* families along with an additional broadleaf is more beneficial than increasing species diversity within a plant family.

The presence of CC reduced weedy species biomass in all CC treatments. Soil building mix and NF were the most effective. This is important in CC performance analysis because as cover crop biomass in SB and NF increases, weed biomass is reduced in an explainable manner. Significant weed biomass reduction through CC incorporation was also observed in other studies [29–31]. Cover crop plants appear to compete well for light and N and suppress weed growth. Consequently, CC can be effective in exhausting the weedy species soil seed bank. Weedy species had the highest plant tissue N concentration when no CC were present. Previous research found that spring CC provided competition to weedy species by scavenging for N and had28% more biomass tissue N than fallow weeds [32]. In this study, MM outcompeted the weedy species for N. This was observed through the elevated C:N ratio and low nitrogen concentration found in the weedy plant tissue. The CC species in the MM treatment were outcompeting the weedy species for soil N and therefore the weeds had less N in the plant tissue, resulting in a higher C:N ratio.

Soil building mix experienced the highest BNF in native soil followed by native soil amended with compost. Legumes present in SB, such as common vetch, were fixing N at a higher rate than other CC. The compost application benefitted MM and increased legume BNF in this mix. Biological N fixation was hindered by the inorganic fertilizer amendment. When inorganic fertilizer was applied in conjunction with cover crops, legume cover crops

**Table 7. Plant total carbon, total nitrogen, carbon to nitrogen ratio (C:N) for cover crops and weeds in cover crop treatments (Control–no cover crop (CON), Phacelia (PH), Soil building mix (SN), Nitrogen fixer mix (NF), and Mycorrhizal mix (MM)).** Means and standard errors followed by different lower-case letters are different among treatments at p ≤ 0.05.

| COVER CROPS | CON | PH | SB | NF | MM | F-Stat | P-Value |
|---|---|---|---|---|---|---|---|
| Total Carbon (%) | 34.3 (3.8) c | 35.33 (0.9) c | 38.4 (2.3) a | 38.1 (2.4) ab | 35.7 (2.0) bc | 6.44 | ≤ 0.01 |
| Total Nitrogen (%) | 2.27 (0.53) | 2.08 (0.35) | 2.04 (0.88) | 2.27 (0.60) | 1.92 (0.40) | 0.1 | n.s. |
| C:N Ratio (nu) | 16 (3) b | 17 (3) ab | 21 (6) a | 18 (3) ab | 19 (4) ab | 3 | ≤ 0.05 |
| **WEEDS** | | | | | | | |
| Total Carbon (%) | 34.3 (3.8) | 34.9 (1.7) | 35.8 (1.4) | 35.9 (2.8) | 35.0 (1.5) | 1.83 | n.s |
| Total Nitrogen (%) | 2.27 (0.53) a | 2.09 (0.34) ab | 2.00 (0.33) ab | 2.02 (0.24) ab | 1.70 (0.43) b | 3.77 | ≤ 0.05 |
| C:N Ratio (nu) | 16 (3) b | 17 (3) b | 18 (3) ab | 18 (2) ab | 22 (5) a | 7 | ≤ 0.01 |

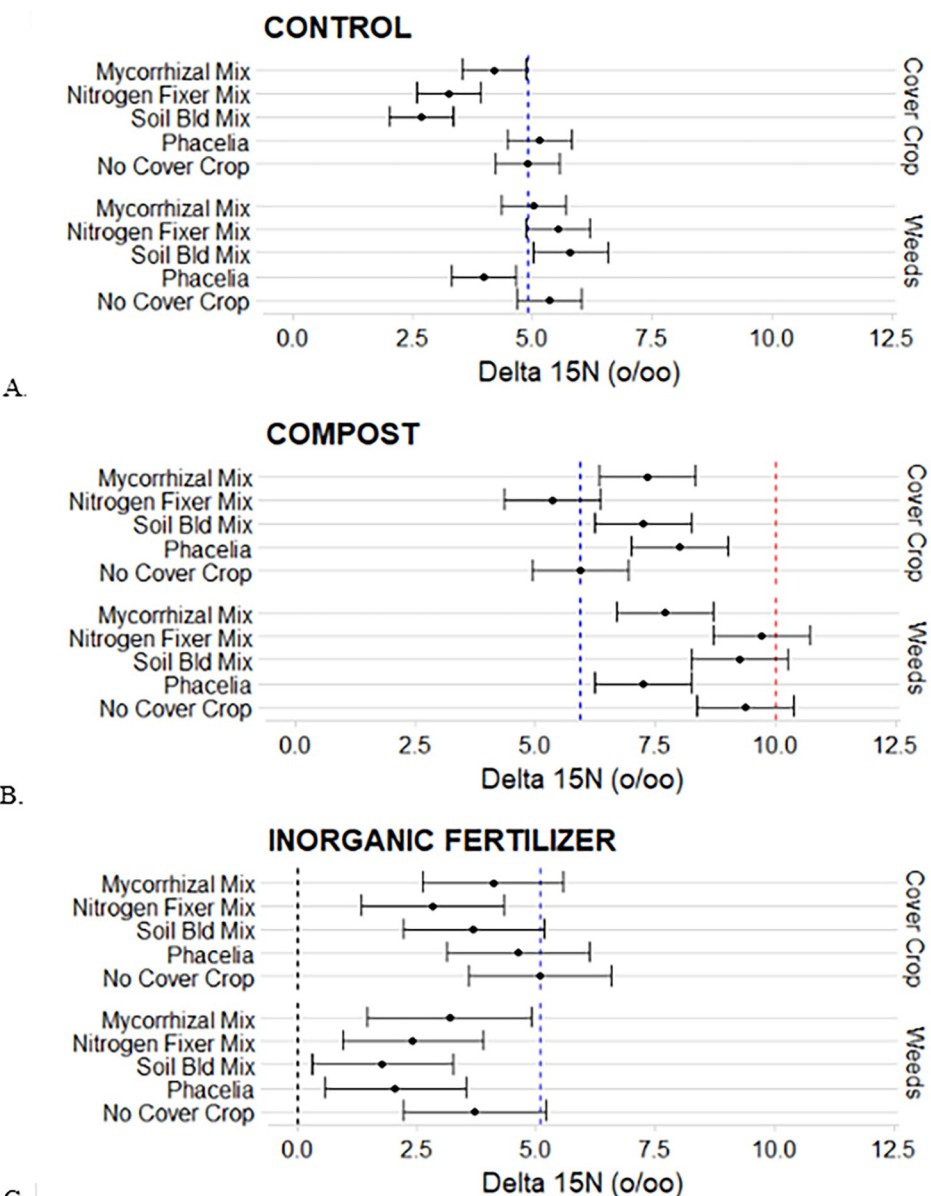

**Fig 4.** Phacelia (PH), Soil building mix (SB), Nitrogen fixer mix (NF), Mycorrhizal mix (MM) and weedy species natural $^{15}$N in soil (A), compost (B), and inorganic fertilizer (C). Blue lines indicate average soil natural $^{15}$N abundance. Red line indicates natural $^{15}$N abundance of the compost and black line indicates average natural $^{15}$N abundance of inorganic fertilizer [28].

utilized N from the fertilizer rather than biologically fixing atmospheric N. When N is readily available from inorganic fertilizer, it is not necessary for CC to establish the symbiotic relationship with rhizobia bacteria. These findings are consistent with other studies where the application of inorganic fertilizer reduced the presence of N fixing bacteria [33].

This study measured and compared $\delta^{15}$N in native soil, soil amended with compost, soil amended with inorganic fertilizer, compost, CC plant tissue and weedy species plant tissue in order to better understand N competition between CC and weedy species. In the control treatment, BNF of legumes present in CC mixes resulted in CC $\delta^{15}$N that fell below the native soil

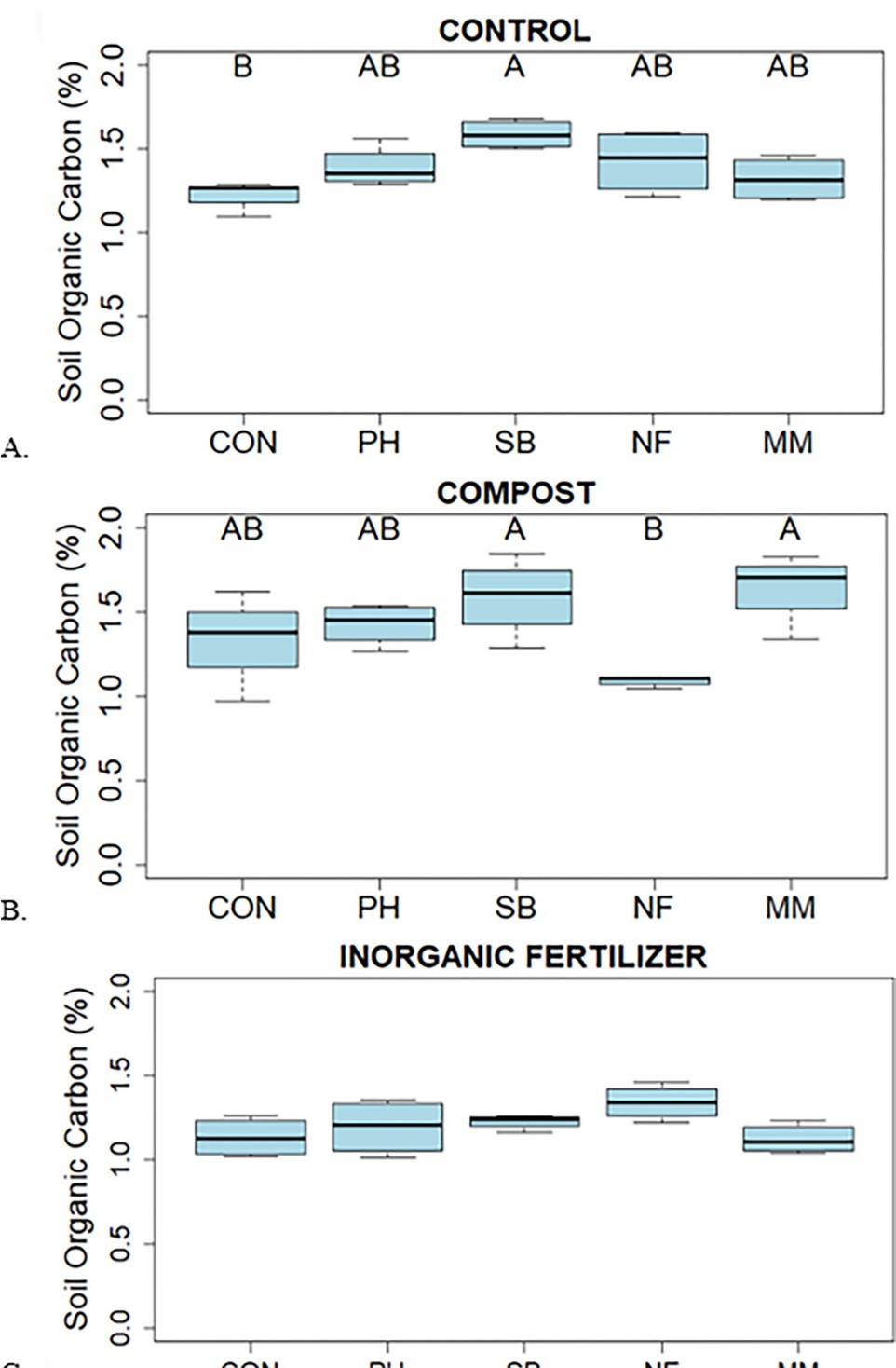

**Fig 5.** Soil organic carbon among cover crop treatments (Control (CON), Phacelia (PH), Soil building mix (SB), Nitrogen fixer mix (NF) and Mycorrhizal mix (MM)) in control (A), compost (B), and inorganic fertilizer (C). Different upper-case letters are different across cc treatments at p ≤ .05.

$\delta^{15}$N. As rhizobia root bacteria begin to fix atmospheric nitrogen and then transfer that nitrogen to the legume plant tissue, the CC $\delta^{15}$N declines towards the air $\delta^{15}$N of 0 ‰. The weedy species $\delta^{15}$N signature was comparable to the native soil $\delta^{15}$N which indicated that weedy species were utilizing soil N. This is important to consider when designing a CC mixture for the purpose of N competition. Cover crop mixes dominated by legume species will not be as competitive with weedy species regarding N accumulation since leguminous CC will fix their own N and weedy species will have access to soil N. Incorporating a Poaceae specie to a CC mixture adds a specie that scavenges for N and competes with weedy species for soil N [32].

Composted cattle manure used in this study had a $\delta^{15}$N of 10.4‰. This is a little lower enrichment of $^{15}$N than reported elsewhere (17.4 ± 1.2‰) [34]. This could be due to the cattle manure used in this study not composting as long as in other studies. The longer a manure composts, the more enriched in $^{15}$N the manure becomes due to $^{14}$N either volatilizing or being utilized by microbes [35,36]. Microbes utilize the $^{14}$N present in the compost as a nitrogen source since it is the lighter isotope which leads to an enrichment of $^{15}$N in the composted cattle manure. Composted cattle manure is left with an enriched $\delta^{15}$N signature when compared to the soil $\delta^{15}$N signature. This enrichment provides a good tracer to see if weedy species and cover crops are utilizing the compost in equal measures.

When native soil was amended with compost that had a $\delta^{15}$N of 10.4 ‰, N uptake from the compost was evidenced in the higher $\delta^{15}$N of both weedy species and CC. Since both weedy species and CC display the higher $\delta^{15}$N, N competition was observed. All weedy species' treatments $\delta^{15}$N increased above the soil $\delta^{15}$Nline and were closer to the compost $\delta^{15}$N. The CC $\delta^{15}$N also increased in most treatments but not all. This is an indication of atmospheric fixation still taking place among legume species in the CC mix. While some CC species were competing for the N found in the compost, legumes in the same CC mix were fixing their own N. In future studies, CC species should be separated into legume and non-legume species prior to isotopic analysis. This would allow for N competition comparisons of non-legume species in a CC mix to weedy species.

The inorganic fertilizer used in this study was manufactured through the Haber Bosch process where atmospheric N is captured and utilized in the manufacturing of inorganic fertilizer. Because of this, the inorganic fertilizer $\delta^{15}$N signal is around 1‰, very similar to the signature of atmospheric N ($\delta^{15}$N of 0‰) [16]. This also gave a credible tracer to observe how CC and weedy species utilized inorganic N. In the Inorganic Fertilizer treatment, the weedy species $\delta^{15}$N fell below the soil $\delta^{15}$N. The CC species $\delta^{15}$N also uniformly decreased signifying inorganic fertilizer N uptake and therefore, competition with weedy species. However, it is hard to tell how much is attributed to inorganic fertilizer and how much is attributed to N fixation since fixing and non-fixing CC species were present in the mixes and not analyzed separately at the time of termination.

In the control treatment, SOC increased when CC were present. Soil building mix had higher SOC in both the control and compost treatments, while MM had higher soil organic C in the compost treatment. This could be since both SB and MM had a higher grass to broadleaf ratio than NF and Phacelia. The fibrous grass roots could provide an increase in rhizodeposits through root exudates which would account for the increase in SOC. In the Inorganic Fertilizer treatment, SOC did not increase across CC treatments. This is consistent with other findings where inclusion of CC increased the SOC stock in cropland soils [37]. In a meta-analysis, Poeplau & Don (2015) found that by incorporating CC, SOC increased at a rate of 0.32 Mg ha$^{-1}$ yr$^{-1}$ to reach a maximum load of 16.7 Mg ha$^{-1}$ [37]. When CC were planted as an intercrop in an almond orchard, SOC increased at a rate of 0.5 Mg ha$^{-1}$ yr$^{-1}$ [38]. This is beneficial for producers regarding soil improvement using CC. This is also beneficial to climatologist as a recommendation for climate mitigation through C capture strategies.

## Conclusions

Soil building mix was the top performing CC. Soil building mix had the lowest planting density among CC mixes but displayed high germination and high biomass production. This is beneficial to the producer looking for good ground coverage on less seed. Soil building mix effectively smothered weeds. Due to the diverse plant families present, this mix displayed beneficial BNF and an increase in SOC. Soil health parameters benefit from each family's contribution. The legumes add BNF while the grasses help increase SOC, benefiting the soil microbes present. Soil building mix outperformed other mixes across all soil and plant parameters. Among CC mixes, SB consisted of the fewest species and had the highest grass to broadleaf ratio. Diversifying plant families is beneficial to a CC mix while increasing number of species in a CC mix does not guarantee better CC performance.

When compost was present, MM performed well in most parameters but only had moderate weed competition. Nitrogen fixer mix had good weed smothering but did not perform as well as in other parameters. Soil building mix and MM were not as heavily dominated with legume species as NF. Incorporating a high performing legume like vetch or clover into a CC mix consisting of a grass and broadleaf seems to be better than incorporating several legume species into a CC mix. With a single, high performing legume, BNF takes place while non legume CC mix species compete with weeds for soil N and improve soil characteristics by increasing SOC.

This research demonstrated promise for cover crops as another tool for producers to utilize for weed competition and soil health improvement. Further research is needed to obtain optimum CC mixes for specific areas. Through proper design, high performing CC species from diverse plant families can be identified. When these species are used together in a CC mix, each specie performs a specific task resulting in agroecosystem benefits.

## Author Contributions

**Conceptualization:** Urszula Norton.

**Data curation:** Elizabeth A. Moore.

**Formal analysis:** Elizabeth A. Moore, Urszula Norton.

**Funding acquisition:** Urszula Norton.

**Visualization:** Elizabeth A. Moore.

**Writing – original draft:** Elizabeth A. Moore, Urszula Norton.

**Writing – review & editing:** Elizabeth A. Moore, Urszula Norton.

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
