## [Decision Letter · Decision Letter 0]

26 Dec 2023

PONE-D-23-38654Improving semi-arid agroecosystem services with cover crop mixesPLOS ONE

Dear Dr. Moore, After careful consideration of your manuscript titled, we appreciate the effort you've invested in the research and the quality of your submission. However, before we can proceed with publication, there are significant revisions required to enhance the clarity, rigor, and overall quality of your work.

We look forward to receiving your revised manuscript.

Kind regards,

Taimoor Hassan Farooq

Academic Editor

PLOS ONE

Journal Requirements:

Reviewers' comments:

Reviewer's Responses to Questions

**Comments to the Author**

1. Is the manuscript technically sound, and do the data support the conclusions?

Reviewer #1: Yes

Reviewer #2: Partly

2. Has the statistical analysis been performed appropriately and rigorously? 

Reviewer #1: Yes

Reviewer #2: No

3. Have the authors made all data underlying the findings in their manuscript fully available?

Reviewer #1: Yes

Reviewer #2: Yes

4. Is the manuscript presented in an intelligible fashion and written in standard English?

Reviewer #1: Yes

Reviewer #2: Yes

5. Review Comments to the Author

Reviewer #1: Subject: Review Report And Suggestions

Manuscript ID “PONE-D-23-38654 ”

“Improving semi-arid agroecosystem services with cover crop mixes”

General Comments.

It is my pleasure to review the manuscript with the ID "Pone-D-23-38654" titled " Improving semi-arid agroecosystem services with cover crop mixes" submitted to the Journal of “PLOS ONE ”There is considerable value in this manuscript in providing insights into the interaction between soil health, cover crops, and soil amendments under low water conditions. This study is clearly explained in terms of experimental design and methodology. A structured presentation is provided for the results and findings. For the manuscript to be deemed well-written, a few technical matters need to be resolved.

I would like to provide my comments and suggestions for major revisions, as well as specific line references.

Technical Comments:

Clarification on Cover Crop Treatments (Lines 55-59):

In the Materials and Methods section, please describe more precisely the composition and purpose of each cover crop treatment (PH, SB, NF, etc.). By comparing the specific characteristics of each cover crop treatment, the reader will be able to better understand the differences between them.

Cover Crop Treatment Effects (Lines 65-68):

Provide an overview of the specific soil health and nutrient competition effects of each cover crop treatment. Analyzing and interpreting the results in more detail will lead to more accurate interpretations.

Temporal Changes in Soil Parameters (Lines 71-78):

The duration of the nine-week study should be discussed in relation to any changes in soil parameters observed over this period. Various factors may be responsible for this, including variations in soil moisture or nutrient levels.

Soil Properties Table (Lines 92-93):

Prepare a short summary of the key soil properties discussed in the manuscript (pH, EC, isotopic signature, etc.). It will be helpful for readers to have a quick reference to this table and will enhance clarity

Clarity on Inorganic Fertilizer Treatment (Lines 121-128):

Explain why inorganic fertilizers were chosen and the impact they were expected to have on soil fertility. Inorganic Fertilizer Treatment Modifications should be understood by the reader from this information.

Data Presentation (Lines 175-184):

When presenting key findings, consider using visual aids like graphs or charts in conjunction with textual descriptions. Providing visual representations of complex data can make it easier for people to understand and access.

Clarify the isotope analysis methodology (Lines 228-236):

This is especially apparent in the calculation of the Biological Nitrogen Fixation (BNF) in

equations (lines 228-229, 231-232). In order to facilitate better understanding, please

provide a brief explanation or reference.

Statistical Analyses (Lines 247-251):

Describe more in detail how the statistical analyses were performed in R. Make sure you

Include a description of the specific tests conducted. Please provide the details on the data

that was used for each analysis, as well as any transformations that have been applied.

Moreover, As a result, statistical approaches will be more transparent due to the availability

of information.

Weed Biomass Regression Analysis (Lines 319-330):

Explain how weed biomass was determined by SB biomass (soil building mix) and NF biomass (nitrogen fixation mix). Provide an explanation of the significance of this analysis in relation to weed suppression. Signatures of 15N isotopes (lines 319-330):

Carbon Sequestration (Lines 428-432):

Discuss the inorganic fertilizer benefit of the cover crop treatments in regard to carbon sequestration. In regard to soil health and carbon capture, discuss the implications of this finding.

Conclusion Section (Lines 440-449):

It would be beneficial to provide more information regarding the superior performance of Soil Building Mix (SB) in comparison to other cover crop mixes. The potential implications of SB's performance for soil improvement and the reasons why SB outperformed others are discussed.

Figures:

Fig 2 (Lines 308-310):

Adding more descriptive labels to the x-axis and y-axis may improve the clarity of the chart. A clear explanation of each treatment abbreviation (CON, PH, SB, NF, MM) should appear in the legend.

Table 7 (Lines 337-339):

To improve the readability of the table, the formatting should be improved. Increase the clarity of the information conveyed in each column by adding more informative column headers.

As a result of addressing these minor suggestions, the manuscript's clarity and overall impact will be further enhanced. A thorough research effort by the authors is commendable, and I look forward to seeing the revised version in the near future.

Regarding my feedback, thank you for taking it into consideration.

Best Regards.

Reviewer #2: The study investigates the effect of different cover crop treatments (species mix for different faring goals) and fertility amendments (control, compost, and mineral fertilizer) on cover crop biomass production and composition, weed suppression, BNF, and soil C sequestration. The nine-week long experiment was conducted under a controlled environment in a greenhouse. I found the statistical data presentation is problematic. There are indications (for e.g., Fig 3) that the two main factors (cover crop and fertility treatments) may have significant interactions. However, the data analysis did not include investigating the interacting effects. The statistical analysis section mentions two-way ANOVA, but data presentation does not reflect that. For example, Fig 2 seems to be averaged across fertility treatments which can only be done if the main effect of fertility and interaction are not significant. Similarly, it is important to show the interaction between cover crop and fertility treatments in Fig 3. The discussion section is weak, sometimes very generic, and mostly focused on BNF. Additionally, I am not sure of using the term “soil C sequestration” for slight increase in soil C for some of the treatments during this 9-week experimental period. This could be a transient increase in active soil C fractions from cover crop rhizodeposits.

Other comments are:

L19-20: Presenting fertilizer rate as N-P-K is more typical than mass of formulation. Also applied to L123-124.

L28: Define “IF”.

L65-68: Complex and unclear. Consider rewriting.

L81: Express it as SI unit

L106: Experimental set up: It seems like compost and inorganic fertilizer were added before cover crop planting. Suh large amount of N and P application during cover crop phase is a concern from nutrient management aspect. Is this the typical practice in NHP wheat rotations with fallow or cover crop? Or fertilizer is generally applied to wheat?

L148: soil water adjustment to 7% was by mass or volume?

L224: you mean legumes absent?

L302-307: Please check the figure reference. It should be Fig 3.

L378-382: This is fundamental knowledge and can be deleted.

Table 7 is not completely visible.

6. PLOS authors have the option to publish the peer review history of their article (what does this mean?). If published, this will include your full peer review and any attached files.

Reviewer #1: **Yes: **Aitezaz Ali Asad Shahani

Reviewer #2: No

---

## [Author Response · Author response to Decision Letter 0]

10 Apr 2024

Response to Reviewers

Manuscript PONE-D-23-38654

Improving semi-arid agroecosystem services with cover crop mixes

Elizabeth A. Moore,1 Urszula Norton,1&2

Authors would like to thank anonymous reviewers for their thoughtful comments and suggestions. Please see below how the comments were resolved.

Reviewer 1 Technical Comments:

With reviewer 1, it seemed like the line numbers given in comments were not consistent with original document, so comments were addressed to the best of my ability. Reviewer 2 comments were consistent with original document.

Clarification on Cover Crop Treatments (Lines 55-59):

In the Materials and Methods section, please describe more precisely the composition and purpose of each cover crop treatment (PH, SB, NF, etc.). By comparing the specific characteristics of each cover crop treatment, the reader will be able to better understand the differences between them. 

Addressed, added additional information in introduction and M&M for each cover crop treatment and beneficial characteristics of plants used in mixes.

Cover Crop Treatment Effects (Lines 65-68):

Provide an overview of the specific soil health and nutrient competition effects of each cover crop treatment. Analyzing and interpreting the results in more detail will lead to more accurate interpretations.

Addressed

Temporal Changes in Soil Parameters (Lines 71-78):

The duration of the nine-week study should be discussed in relation to any changes in soil parameters observed over this period. Various factors may be responsible for this, including variations in soil moisture or nutrient levels.

Addressed

Soil Properties Table (Lines 92-93):

Prepare a short summary of the key soil properties discussed in the manuscript (pH, EC, isotopic signature, etc.). It will be helpful for readers to have a quick reference to this table and will enhance clarity.

Addressed, all requested information is found in Table 1 and discussed in lines 128-145.

Clarity on Inorganic Fertilizer Treatment (Lines 121-128):

Explain why inorganic fertilizers were chosen and the impact they were expected to have on soil fertility. Inorganic Fertilizer Treatment Modifications should be understood by the reader from this information.

Addressed, additional information was provided in M&M (lines 175-186).

Data Presentation (Lines 175-184):

When presenting key findings, consider using visual aids like graphs or charts in conjunction with textual descriptions. Providing visual representations of complex data can make it easier for people to understand and access.

Experimental layout is included in separate tif file.

Clarify the isotope analysis methodology (Lines 228-236):

This is especially apparent in the calculation of the Biological Nitrogen Fixation (BNF) in

equations (lines 228-229, 231-232). In order to facilitate better understanding, please

provide a brief explanation or reference.

Addressed, additional information added to introduction and M&M.

Statistical Analyses (Lines 247-251):

Describe more in detail how the statistical analyses were performed in R. Make sure you

Include a description of the specific tests conducted. Please provide the details on the data

that was used for each analysis, as well as any transformations that have been applied.

Moreover, As a result, statistical approaches will be more transparent due to the availability

of information.

Addressed in M&M.

Weed Biomass Regression Analysis (Lines 319-330):

Explain how weed biomass was determined by SB biomass (soil building mix) and NF biomass (nitrogen fixation mix). Provide an explanation of the significance of this analysis in relation to weed suppression. Addressed in M&M and in discussion.

Carbon Sequestration (Lines 428-432):

Discuss the inorganic fertilizer benefit of the cover crop treatments in regard to carbon sequestration. In regard to soil health and carbon capture, discuss the implications of this finding.

Carbon sequestration term was removed and replaced with increase of soil organic carbon. This was addressed in discussions section.

Conclusion Section (Lines 440-449):

It would be beneficial to provide more information regarding the superior performance of Soil Building Mix (SB) in comparison to other cover crop mixes. The potential implications of SB's performance for soil improvement and the reasons why SB outperformed others are discussed.

Addressed in conclusions section.

Figures:

Fig 2 (Lines 308-310):

Adding more descriptive labels to the x-axis and y-axis may improve the clarity of the chart. A clear explanation of each treatment abbreviation (CON, PH, SB, NF, MM) should appear in the legend.

Addressed except for legend. An explanation for CC treatment abbreviations is in the figure description. 

Table 7 (Lines 337-339):

To improve the readability of the table, the formatting should be improved. Increase the clarity of the information conveyed in each column by adding more informative column headers.

Addressed, Table 7 was removed due to duplicate information presented in Fig. 4

Reviewer 2 Technical Comments

I found the statistical data presentation is problematic. There are indications (for e.g., Fig 3) that the two main factors (cover crop and fertility treatments) may have significant interactions. However, the data analysis did not include investigating the interacting effects. The statistical analysis section mentions two-way ANOVA, but data presentation does not reflect that. 

Tests for interactions were conducted on cover crop treatment and soil amendment and no interactions were found. 

For example, Fig 2 seems to be averaged across fertility treatments which can only be done if the main effect of fertility and interaction are not significant.

Tests for interactions were conducted on cover crop treatment and soil amendment were conducted and no interactions were found. 

Similarly, it is important to show the interaction between cover crop and fertility treatments in Fig 3.

No interactions were found. 

The discussion section is weak, sometimes very generic, and mostly focused on BNF. 

Addressed, additional discussion was provided.

Additionally, I am not sure of using the term “soil C sequestration” for slight increase in soil C for some of the treatments during this 9-week experimental period. This could be a transient increase in active soil C fractions from cover crop rhizodeposits.

Valid comment, addressed, verbiage changed to increase in soil organic carbon rather than sequestration.

L19-20: Presenting fertilizer rate as N-P-K is more typical than mass of formulation. Also applied to L123-124.

Addressed

L28: Define “IF”. 

Addressed 

L65-68: Complex and unclear. Consider rewriting. 

Addressed

L81: Express it as SI unit 

Addressed

L106: Experimental set up: It seems like compost and inorganic fertilizer were added before cover crop planting. Such large amount of N and P application during cover crop phase is a concern from nutrient management aspect. Is this the typical practice in NHP wheat rotations with fallow or cover crop? Or fertilizer is generally applied to wheat?

Addressed

L148: soil water adjustment to 7% was by mass or volume? 

By mass, addressed

L224: you mean legumes absent? 

Yes, good catch, addressed

L302-307: Please check the figure reference. It should be Fig 3. 

Good catch, addressed

L378-382: This is fundamental knowledge and can be deleted.

Addressed

Table 7 is not completely visible. 

This table has been deleted due to duplicate information being presented in Fig. 4

---

## [Decision Letter · Decision Letter 1]

19 Jun 2024

Improving semi-arid agroecosystem services with cover crop mixes

PONE-D-23-38654R1

Dear authors,

We’re pleased to inform you that your manuscript has been judged scientifically suitable for publication and will be formally accepted for publication once it meets all outstanding technical requirements.

Kind regards,

Taimoor Hassan Farooq

Academic Editor

PLOS ONE

Additional Editor Comments (optional):

We are pleased to inform you that your paper has been accepted for publication. After thorough review, the reviewers have provided positive feedback, highlighting the significant contributions your work makes to the field. Congratulations!

Reviewers' comments:

Reviewer's Responses to Questions

**Comments to the Author**

1. If the authors have adequately addressed your comments raised in a previous round of review and you feel that this manuscript is now acceptable for publication, you may indicate that here to bypass the “Comments to the Author” section, enter your conflict of interest statement in the “Confidential to Editor” section, and submit your "Accept" recommendation.

Reviewer #3: All comments have been addressed

2. Is the manuscript technically sound, and do the data support the conclusions?

Reviewer #3: Yes

3. Has the statistical analysis been performed appropriately and rigorously? 

Reviewer #3: Yes

4. Have the authors made all data underlying the findings in their manuscript fully available?

Reviewer #3: Yes

5. Is the manuscript presented in an intelligible fashion and written in standard English?

Reviewer #3: Yes

6. Review Comments to the Author

Reviewer #3: (No Response)

7. PLOS authors have the option to publish the peer review history of their article (what does this mean?). If published, this will include your full peer review and any attached files.

Reviewer #3: No

---

## [Editor Report · Acceptance letter]

24 Jun 2024

PONE-D-23-38654R1 

PLOS ONE

Dear Dr. Moore, 

I'm pleased to inform you that your manuscript has been deemed suitable for publication in PLOS ONE. Congratulations! Your manuscript is now being handed over to our production team.

Kind regards, 

on behalf of

Taimoor Hassan Farooq 

Academic Editor

PLOS ONE